# Role of Daratumumab, Lenalidomide, and Dexamethasone in Transplantation-Eligible Patients with Multiple Myeloma After the Failure of Bortezomib-Based Induction Therapy [note 1]

**DOI:** 10.3390/hematolrep17060057

**Published:** 2025-10-29

**Authors:** Shun Ito, Takashi Hamada, Masaru Nakagawa, Takashi Ichinohe, Hironao Nukariya, Toshihide Endo, Kazuya Kurihara, Yuichi Takeuchi, Shimon Otake, Hiromichi Takahashi, Hideki Nakamura, Katsuhiro Miura

**Affiliations:** 1Division of Hematology and Rheumatology, Department of Medicine, Nihon University School of Medicine, Tokyo 173-8610, Japan; ito.shun@nihon-u.ac.jp (S.I.); hamada.takashi@nihon-u.ac.jp (T.H.); nakam576@kasukabe-city-hosp.jp (M.N.); ichinohe.takashi@nihon-u.ac.jp (T.I.); nukariya.hironao@nihon-u.ac.jp (H.N.); endo.toshihide@nihon-u.ac.jp (T.E.); kurihara.kazuya@nihon-u.ac.jp (K.K.); takeuchi.yuuichi@nihon-u.ac.jp (Y.T.); ootake.shimon@nihon-u.ac.jp (S.O.); takahashi.hiromichi@nihon-u.ac.jp (H.T.); nakamura.hideki@nihon-u.ac.jp (H.N.); 2Division of Microbiology, Department of Pathology and Microbiology, Nihon University School of Medicine, Tokyo 173-8610, Japan; 3Department of Hematology, Kasukabe Medical Center, Kasukabe 344-8588, Saitama, Japan

**Keywords:** autologous stem cell transplantation, bortezomib, cyclophosphamide, daratumumab, lenalidomide, multiple myeloma

## Abstract

**Background/Objectives:** The role of daratumumab, lenalidomide, and dexamethasone (DRd) in autologous stem cell transplantation (ASCT)-eligible patients with multiple myeloma (MM) after first-line bortezomib, cyclophosphamide, and dexamethasone (VCd) treatment is not yet established. **Methods:** We retrospectively evaluated ASCT-eligible patients with MM who received second-line therapy with DRd after initial induction therapy with VCd between 2017 and 2023 (salvage group). For comparison, patients who successfully underwent per-protocol treatment with VCd induction, followed by ASCT during the same period, were selected (control group). **Results:** Eight patients with a median age of 61 years (range, 36–68 years) were included in the salvage group. After a median of 5 DRd cycles, the best response was partial response (PR) in two patients (25%) and a very good partial response (VGPR) in six (75%). All patients underwent ASCT, resulting in PR in one (13%), VGPR in four (50%), and stringent complete response in three (38%). Measurable residual disease (MRD) assessed using multicolor flow cytometry was negative in four patients (50%). The controls included thirteen patients with a median age of 60 years (range, 44–64 years). While most patients in both groups received various post-ASCT therapies, the post-ASCT 2-year time to the next treatment rate was slightly better in the salvage group than in the control group (88% vs. 49%, *p* = 0.089). However, hypogammaglobulinemia was more common in the salvage group (75% vs. 15%, *p* = 0.018). **Conclusions:** This small case series suggests that DRd is promising for ASCT-eligible patients with MM after VCd failure.

## 1. Introduction

Treatment of multiple myeloma (MM) has advanced with the development of novel agents, including proteasome inhibitors, immunomodulatory drugs, and anti-CD38 antibodies, resulting in significant improvements in the long-term prognosis of patients with MM. Nonetheless, autologous stem cell transplantation (ASCT) continues to play a crucial role in the treatment of transplant-eligible patients with newly diagnosed MM (NDMM) [1]. Particularly, bortezomib, cyclophosphamide, and dexamethasone (VCd) therapy has been widely used as the standard induction regimen for transplant-eligible patients with NDMM since around 2009 [2,3]. Recently, bortezomib, lenalidomide, and dexamethasone (VRd) therapy has also become the preferred induction regimen because of its reported efficacy [4,5].

Daratumumab is a monoclonal antibody that targets CD38, a glycoprotein widely expressed in normal and malignant plasma cells. It induces the lysis of MM cells through Fc-mediated crosslinking, triggering direct apoptosis, as well as antibody-dependent cellular cytotoxicity and complement-dependent cytotoxicity [6]. Furthermore, daratumumab promotes T-cell expansion by depleting CD38 on non-plasma regulatory immune cells [7]. Such an action on the tumor microenvironment is expected to work synergistically with lenalidomide, a representative immunomodulatory drug with strong anti-tumor effects on MM [8].

The POLLUX trial demonstrated that adding daratumumab to lenalidomide and dexamethasone (DRd) significantly improved progression-free survival (PFS) and overall survival (OS) in patients with relapsed or refractory multiple myeloma (RRMM) compared with lenalidomide and dexamethasone alone(Rd) [9]. Subsequently, the MAIA study showed superior clinical outcomes of DRd compared with Rd in ASCT-ineligible patients with NDMM [10]. Consequently, DRd has become the standard treatment for RRMM and NDMM. However, because its efficacy has primarily been evaluated in ASCT-ineligible settings, its role among ASCT-eligible patients with MM remains unclear. Therefore, we conducted a case series study on ASCT-eligible patients who received second-line treatment with DRd after failing to respond to induction therapy with VCd to determine if this second-line treatment could be a potential option in cases where an optimal response to VCd induction therapy was not achieved or when patients were unable to tolerate the treatment.

## 2. Materials and Methods

This retrospective study included transplant-eligible patients with NDMM who underwent VCd induction at our institution from November 2017, when DRd became available for RRMM in Japan, to April 2023. During this period, the VCd regimen was employed as the initial induction therapy for ASCT-eligible patients with NDMM in our institution. Thus, patients were included in the study if they underwent initial induction therapy with VCd and attempted to receive autologous stem cell transplantation (ASCT) after achieving at least a partial response (PR), as per our institution’s protocol at the time. Among them, we focused on patients who received DRd as second-line therapy after failing to respond to the initial induction therapy with VCd (salvage group). Furthermore, we selected patients who successfully underwent per-protocol treatment with VCd induction, followed by ASCT (control group) for comparison. Accordingly, we excluded patients who died during VCd induction, refused to undergo ASCT, and received second-line therapies other than DRd.

The primary measurements were the response to second-line DRd induction and flow cytometry-assessed measurable residual disease (flow-MRD) status at any time point in the salvage group. Other measurements included response after ASCT, post-ASCT flow-MRD negative rates, post-ASCT 2-year time to next treatment (TTNT), post-ASCT 2-year OS, and toxicities in both groups. For the response evaluation, we used “flexible” response criteria, which modified the International Myeloma Working Group response criteria to adopt the real-world practice, where repeated bone marrow samplings are occasionally lacking [11,12]. We assessed flow-MRD using a 10-color multiparameter flow cytometry analysis of bone marrow specimens, which significantly correlates with the EuroFlow next-generation flow, performed by BML Inc. (Tokyo, Japan) [13]. The assay used antibodies including CD38 multiepitope FITC (polyclonal, CYT-38F2) (Cytognos, Salamanca, Spain), CD138 V450 (MI15), CD45 V500-C (2D1), CD19 APC-H7 (SJ25C1), CD56 PE (MY31), CD27 APC-R700 (M-T271), CD81 BV605 (JS-81), CD117 PE-Cy7 (104D2) (BD Biosciences, Milpitas, CA, USA), cytoplasmic immunoglobulin (cIg) κAPC (polyclonal, PR712), and cIgλ PerCP-Cy5.5 (polyclonal, C0222) (Dako, Glostrup, Denmark). Data acquisition and analysis were performed using FACSLyric and FACSuite (BD Biosciences), respectively [13].

We conducted statistical analyses using JMP Pro version 18 software (SAS Institute, Cary, NC, USA). Continuous and categorical variables were summarized as medians with ranges (minimum–maximum) and absolute numbers and percentages (%), respectively. The statistical significance of continuous variables was evaluated using the Mann–Whitney U test; however, Fisher’s exact test was used to compare categorical variables. The 2-year TTNT and OS rates were estimated using the Kaplan–Meier method, and the log-rank test was used for intergroup comparisons. Each test was two-sided, and statistical significance was set at *p* < 0.05.

## 3. Results

During the study period, we treated eight patients with DRd as second-line therapy after the initial induction therapy with VCd (Figure 1). The median age was 61 years (range, 36–68 years, with six men and two women. The revised International Staging System stages at initial diagnosis were I, II, and III in one, two, and three patients, respectively [14]. Among them, the responses to VCd were progressive disease (PD) in one patient (13%), stable disease in four (50%), and PR in three patients (38%). The reasons for switching to DRd in the three patients who achieved PR to VCd were that the serum M-protein level reached a plateau at a high level in two patients, and the other patient was intolerant owing to nausea and peripheral neuropathy.

DRd was administered for a median of 5 (range, 3–7) cycles before ASCT, and the best response to DRd was a PR in two patients (25%) and a very good partial response (VGPR) in six patients (75%). During or after the re-induction therapy with DRd, all patients proceeded to peripheral blood stem cell collection using pegylated granulocyte-colony stimulating factor and plerixafor, resulting in a median of 2.4 (range, 1.60–5.19) × 10^6^/Kg CD34-positive cells harvested. Among them, seven patients subsequently underwent high-dose melphalan therapy followed by ASCT. One patient experienced serum M-protein elevation in a later cycle of DRd after stem cell collection and received third-line therapy with carfilzomib and dexamethasone for four cycles. Subsequently, the patient achieved PR and received high-dose melphalan, followed by ASCT. The results of their post-ASCT response evaluation were PR in one (13%), VGPR in three (38%), stringent complete response (sCR) in four (50%), and flow-MRD negative in four patients (50%).

Post-ASCT therapy included DRd in five patients, Rd in one, elotuzumab, pomalidomide, and dexamethasone in one, and no treatment in one patient. At the time of analysis, seven patients continued ongoing treatment with a median remission duration of 30 months (range, 18–62 months), but one patient experienced PD at 12 months, followed by the next treatment. Two additional patients converted to flow-MRD-negative status during post-ASCT therapy with DRd (Figure 2).

Thirteen patients were included in the control group (Figure 1). The median age was 60 years (range, 44–64), including six men and seven women. Responses to VCd induction were PR in four patients (31%), VGPR in seven (54%), and sCR in two (15%). Their median CD34-positive cell count, obtained using the same procedure as the control group, was 5.3 × 10^6^/Kg (range, 1.92–13.1), which was significantly higher than that of the salvage group (*p* = 0.013). The post-ASCT treatments consisted of Rd in six patients, ixazomib in five, and no treatment in two. A comparison of the clinical characteristics of the control and salvage groups is summarized in Table 1.

As a result, sCR and flow-MRD negative rates after ASCT were similar in the salvage and control groups (50% vs. 54%, *p* = 1.000; and 38% vs. 38%, *p* = 1.000, respectively). A landmark analysis of post-ASCT 2-year TTNT in the salvage and control groups revealed 88% and 52%, respectively (*p* = 0.089). The post-ASCT 2-year OS in these groups was 88% and 92%, respectively (*p* = 0.708) (Figure 3). The toxicity profile was generally favorable in both groups; however, hypogammaglobulinemia (serum IgG level < 500 mg/dL) was significantly more frequent in the salvage group than in the control group (75% vs. 15%, *p* = 0.018). However, this was not associated with infectious episodes ≥ grade 3, partly because of the use of appropriate immunoglobulin supplementation.

## 4. Discussion

In this study, we illustrated that second-line therapy with DRd, followed by ASCT with or without post-ASCT consolidation/maintenance therapies, represents a promising option for ASCT-eligible patients with MM who did not achieve an optimal response to VCd induction therapy or were unable to tolerate it. Patients in the salvage group demonstrated excellent clinical outcomes, with an overall response and VGPR rate of 100% (8/8) and 75% (6/8), respectively, following reinduction with DRd. Before the daratumumab era, primary failure of bortezomib-based induction therapies was a significant predictor of poor survival in NDMM [15]. However, the salvage group exhibited outcomes similar to those of the control group, which was a highly selected cohort expected to have a favorable prognosis. The post-ASCT 2-year TTNT rate was higher in the salvage group than in the control group, despite marginal statistical significance (*p* = 0.089).

It is noteworthy that the administration of post-ASCT DRd led to MRD negativity in two additional patients, suggesting that prolonged DRd therapy may enhance the treatment response and contribute to sustained disease control. This finding is consistent with that of the AURIGA trial, which demonstrated that daratumumab-based maintenance therapy significantly improved MRD negativity rates and PFS compared with lenalidomide monotherapy [16]. MRD has become a key prognostic marker in MM, and recent clinical trials have used it as a surrogate endpoint based on its predictive value for long-term clinical outcomes [17,18]. In this context, the finding that 75% (6/8) of the patients in the salvage group ultimately achieved flow-MRD negativity is notable.

It is also important to note the disadvantages of DRd salvage after VCd treatment. Regarding hematopoietic stem cell collection, the salvage group had fewer CD34-positive cells than the control group. In cases of 1.6–2.0 × 10^6^/Kg CD34-positive cells, we carefully performed ASCT after shared decision-making with the patients and their families [19]. Therefore, prolonged cycles of second-line DRd should be avoided before harvesting stem cells. A recent meta-analysis also indicated a negative impact of prior lenalidomide or daratumumab exposure on hematopoietic stem cell mobilization or engraftment, suggesting a possible role for plerixafor in mitigating this effect [20]. Given these findings, using plerixafor for stem cell collection after short courses of salvage DRd may be a reasonable approach for patients planning to undergo ASCT. Regarding toxicity, the incidence of hypogammaglobulinemia was significantly higher in the salvage group but was not related to grade 3 or higher infections. In contrast, the POLLUX trial reported a higher incidence of grade 3–4 infections with DRd than with Rd therapy (28% vs. 23%) [9]. This discrepancy may be because of the involvement of ASCT-ineligible patients with POLLUX, who typically have different baseline characteristics. However, long-term follow-up of POLLUX indicated that the incidence of grade 3–4 infections was 45% in the DRd group, highlighting the increased risk of infection associated with extended DRd therapy [21]. Therefore, further investigations are needed to establish the optimal treatment duration for DRd and effective infection control. For instance, immunoglobulin replacement may be considered for patients showing hypogammaglobulinemia with a serum IgG concentration below 400 mg/dL or those experiencing recurrent bacterial infections, as increasing evidence supports its clinical benefit for such high-risk patients with MM [22]. Accordingly, monitoring serum IgG levels is crucial in patients undergoing salvage and post-ASCT treatment with DRd.

Lim et al. recently reported the results of the ALLG MM21 trial, conducted in Australia [23]. They performed four-cycle DRd salvage, high-dose melphalan-based ASCT, 12-cycle DRd consolidation, and R maintenance until PD or unacceptable toxicities for 50 ASCT-eligible patients with MM who did not respond optimally to first-line VCd induction. The treatment resulted in an overall response rate of 72% for the DRd salvage, with Euroflow MRD negativity in 46% and 79% of the intention-to-treat population and evaluable patients, respectively, after the DRd consolidation. It achieved 73% 1.5-year PFS and 86% 1.5-year OS, accompanied by a favorable safety profile [23]. The patient selection and response evaluation were less strict in our study; however, our findings confirmed that this strategy can be effectively applied in clinical practice. Additionally, we identified the risk of poor hematopoietic stem cell mobilization and hypogammaglobulinemia in this population. These eye-opening findings were based on a comparison with a cohort well-treated with VCd, which was not examined in their trial.

Global trends in the induction of ASCT-eligible NDMM have changed over the past decade. Whereas no direct comparisons have been performed, VRd has been increasingly adopted because of its potential effect on deeper responses [24]. Additionally, a recent study showed positive outcomes with a four-drug induction therapy, including daratumumab, bortezomib, lenalidomide, and dexamethasone, followed by ASCT, and maintenance therapy with daratumumab and lenalidomide [25]. However, VCd remains a valuable treatment option, particularly for patients with impaired renal function owing to its favorable renal toxicity profile [26].

This study provides empirical evidence supporting the efficacy of DRd in pre- and post-ASCT settings; however, it has some limitations. First, because this was a small case series from a single institution, selection, information, survivor, and other biases were unavoidable. Particularly, the small sample size of the salvage group warrants further investigation on the role of DRd in this population. Also, a selection bias in the control group might affect study results. Nonetheless, our findings are comparable to those of a well-designed clinical trial conducted by Lim et al., suggesting that our results may be reproducible in other contexts. Second, the treatment protocol was not entirely consistent, particularly post-ASCT treatments. This variability was because of the choice of maintenance or consolidation therapy based on the patient’s preference and the physician’s discretion. Growing evidence may enhance more sophisticated post-ASCT strategies for patients in this setting [27,28,29,30]. Finally, the bone marrow and imaging studies for response evaluation at each time point were not fully organized, raising the possibility that we overestimated the therapeutic effects in both groups. In addition, applying TTNT, not PFS, for the study endpoint is somewhat arbitrary. This is because patients occasionally received the next treatment after paraprotein relapse, which did not meet the criteria for PD. This substitution may also have also introduced a physician bias. However, we performed intermittent flow-MRD assays in most cases, which helped ensure an accurate assessment of remission status.

In conclusion, second-line DRd is a promising option for ASCT-eligible patients with MM who do not respond optimally to first-line induction therapy with VCd. Although VCd induction is used less frequently, our findings are valuable for selected patients.

## Figures and Tables

**Figure 1 hematolrep-17-00057-f001:**
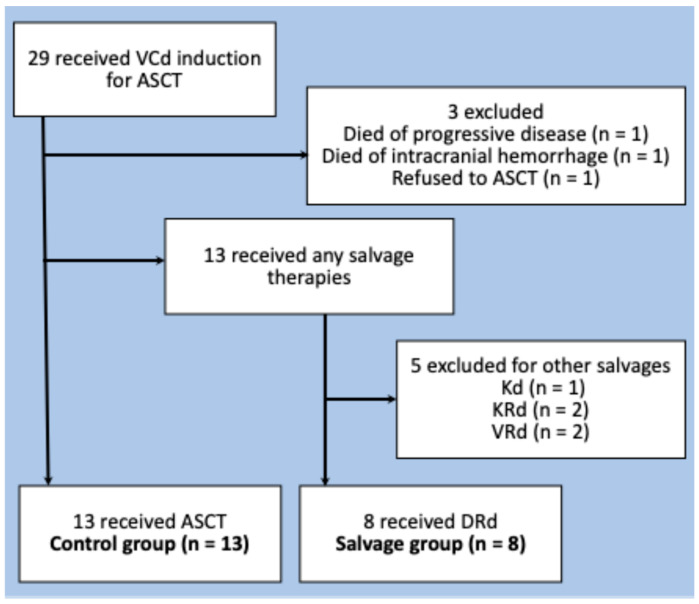
Patient selection. Out of the 29 patients who received initial induction therapy with bortezomib, cyclophosphamide, and dexamethasone (VCd) for autologous stem cell transplantation (ASCT), 8 were assigned to the salvage group and 13 to the control group. Among the 13 patients who underwent salvage therapies, 5 were excluded because they received treatments other than VCd, such as carfilzomib and dexamethasone (Kd), carfilzomib, lenalidomide, and dexamethasone (KRd), or bortezomib, lenalidomide, and dexamethasone (VRd).

**Figure 2 hematolrep-17-00057-f002:**
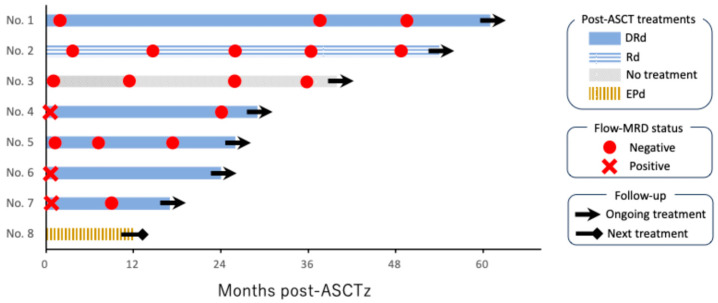
Swimmer plot showing post-ASCT clinical course in eight patients. “No.” denotes the patient number, and they are listed in order by the duration of post-autologous stem cell transplantation (ASCT) treatment. Four patients achieved flow cytometry-assessed measurable residual disease (flow-MRD) negative status directly after ASCT. Two additional patients achieved flow-MRD negative status following post-ASCT treatment with daratumumab, lenalidomide, and dexamethasone (DRd). Only one patient, who received elotuzumab, pomalidomide, and dexamethasone (EPd), proceeded to the next treatment due to disease progression.

**Figure 3 hematolrep-17-00057-f003:**
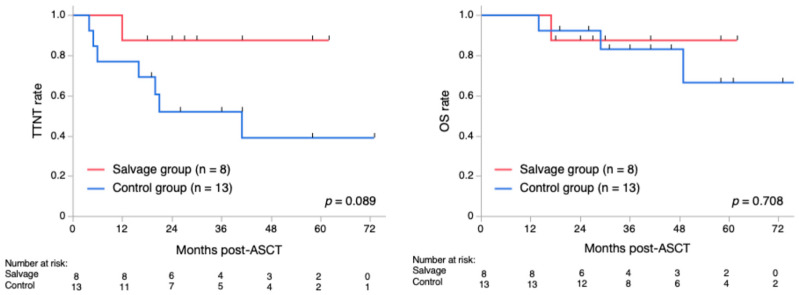
Time to next treatment and overall survival after ASCT. The red and blue lines represent the salvage and control groups, respectively. The 2-year time to next treatment (TTNT) after autologous stem cell transplantation (ASCT) was slightly better in the salvage group than in the control group, with marginal statistical significance (88% vs. 52%, *p* = 0.089). The 2-year overall survival (OS) after ASCT was similar between the groups (88% vs. 92%, *p* = 0.708).

**Table 1 hematolrep-17-00057-t001:** Clinical characteristics of patients.

Items	Salvage Group (*n* = 8)	Control Group (*n* = 13)	*p* Value
Men/women, *n*	6/2	6/7	0.367
Median age, y (range)	61 (36–68)	60 (44–64)	0.828
R-ISS stage, *n* (%)			1.000
Stage I or II	5 (63%)	9 (69%)	
Stage III	3 (38%)	4 (31%)	
Performance status, *n* (%)			0.618
0–1	6 (75%)	11 (85%)	
≥2	2 (25%)	2 (15%)	
High-risk karyotype *, *n* (%)			1.000
Present	5 (63%)	7 (54%)	
Absent	3 (38%)	6 (46%)	
Bone lysis, *n* (%)			0.618
Present	6 (75%)	11 (85%)	
Absent	2 (25%)	2 (15%)	
Extramedullary involvement, *n* (%)			
Present	2 (25%)	4 (31%)	1.000
Absent	6 (75%)	9 (69%)	
Best response to (re)induction, *n* (%)			1.000
PR	2 (25%)	4 (31%)	
≥VGPR	6 (75%)	9 (69%)	
Post-ASCT treatment, *n* (%)			1.000
Present	7 (88%)	11 (85%)	
Absent	1 (13%)	2 (15%)	

R-ISS, Revised International Staging System; ASCT, autologous stem cell transplantation. * At least one of the following karyotypes was identified: del(17p), t(4;14), t(11;14), t(14;16), or 1q21+.

## Data Availability

Anonymized data from this study may be provided upon reasonable request to the corresponding author.

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
