# Peer review of "Role of Daratumumab, Lenalidomide, and Dexamethasone in Transplantation-Eligible Patients with Multiple Myeloma After the Failure of Bortezomib-Based Induction Therapyâ€"

_hematolrep, 2025, doi:10.3390/hematolrep17060057_

Round 1
Reviewer 1 Report
The authors should aim to present their findings using a formal scientific style, which will significantly enhance the quality and impact of the manuscript.
The manuscript presents the results on the role of daratumumab–lenalidomide–dexamethasone in transplantation-eligible patients with multiple myeloma after the failure of bortezomib-based induction therapy. It is an interesting study, however there are several issues that should be addressed:
- Introduction: The authors should expand the Introduction to include relevant biology on the therapeutic agents used. Specifically, a brief overview of the mechanisms of action of daratumumab, lenalidomide, and dexamethasone would provide essential context for the study.
- Methods and Results: All clinical and baseline patient characteristics currently presented in the Results section should be presented in the Methods section. The Results section should only include outcomes of the study.
- Flow Cytometry: The Methods section should include a detailed description of the flow cytometry panel used. This should cover the list of antibodies, catalog numbers, suppliers, the model of the flow cytometer, and the analysis software employed.
- Figures: All figures should include comprehensive and self-explanatory legends.
Author Response
Major Comments
The authors should aim to present their findings using a formal scientific style, which will significantly enhance the quality and impact of the manuscript.
Response: Thank you so much for your advice. We have duly addressed the reviewer's helpful comments.
1. Introduction: The authors should expand the Introduction to include relevant biology on the therapeutic agents used. Specifically, a brief overview of the mechanisms of action of daratumumab, lenalidomide, and dexamethasone would provide essential context for the study.
Response: We appreciate this important suggestion. We inserted a brief overview of the mechanisms of action and the potential synergistic effect of daratumumab and lenalidomide in the Introduction (Lines 57–63).
2. Methods and Results: All clinical and baseline patient characteristics currently presented in the Results section should be presented in the Methods section. The Results section should only include outcomes of the study.
Response: We appreciate this helpful suggestion. In fact, we did not fully identify the patients eligible for analysis when designing this retrospective study. During the study period, we employed the VCd regimen as initial induction therapy for ASCT-eligible patients with MM. The study was started by identifying eligible patients from the database. Thus, we outlined the patient selection procedure in the Materials and Methods section and described the identified patients and their clinical characteristics in the Results section, as is typically done in this type of clinical report. For a better understanding of readers, we have added a description of this study process at Lines 81–83. We sincerely request that the reviewer accept that we retain the patient characteristics in the Results section, as these are the exact results of the study protocol.
3. Flow Cytometry: The Methods section should include a detailed description of the flow cytometry panel used. This should cover the list of antibodies, catalog numbers, suppliers, the model of the flow cytometer, and the analysis software employed.
Response: We appreciate this important suggestion. We have included details of the antibody, reagents, manufacturers, and clones, as well as the instrumentation and its software of the multicolour flow cytometer (Lines 102–108).
4. Figures: All figures should include comprehensive and self-explanatory legends.
Response: We appreciate this important suggestion. We have included comprehensive and self-explanatory legends in each figure.
Reviewer 2 Report
- The sample size of salvage group is very low (n = 8). This limitation should be highlighted and considered int he discussion as well as in abstract.
- PFS (progression free survival) is a more conventional endpoint that TTNT. The rationale for considering TTNT should be discussed and the potential of bias that could arise should also be considered and mentioned.
- Selection bias could arise for the control group being "highly selected". This can be addresed in the discussion to make sure the results are interpreted correctly.
- The finding of lower CD34+ cell yields in the salvage group is important. The manuscript should discuss whether DRD or lenalidomide specifically contributed to poor mobilization, and whether alternative mobilization strategies were considered.
- The higher rate of hypogammaglobulinemia in the salvage group is notable, but the manuscript should provide more detail on infection monitoring, prophylaxis, and management, especially given the known infection risks with daratumumab.
Need more clarity on figures and tables:
-
Figure 1:
Please label exclusion criteria and group assignments more clearly in the flow diagram, and indicate the number of patients at each step and in the final analysis. -
Figure 2:
The legend should define all symbols and clarify abbreviations such as “EPd” and patient numbers. -
Figure 3 :
Axes and group labels need clearer definition. Specify in the legend which line represents each group, and, if possible, add the number at risk at each time point.
Ensure all abbreviations (e.g., sCR, VGPR, MRD, TTNT, EPd) are defined at first use in both the main text and figure/table legends. For example, “EPd” is used in Figure 2 but not defined in the abbreviations list.
Consistently use standard abbreviations (e.g., sCR, VGPR, MRD) and define them at first use.
Author Response
Major Comments
1. The sample size of salvage group is very low (n = 8). This limitation should be highlighted and considered in the discussion as well as in abstract.
Response: We appreciate this important suggestion. We have addressed this most significant limitation of our study in the abstract (Lines 39–40) and the Discussion section (Lines 261–262).
2. PFS (progression free survival) is a more conventional endpoint that TTNT. The rationale for considering TTNT should be discussed and the potential of bias that could arise should also be considered and mentioned.
Response: We appreciate this valuable suggestion. We have addressed a potential physician bias related to substituting PFS with TTNT in the Discussion section (Line 274).
3. Selection bias could arise for the control group being "highly selected". This can be addresed in the discussion to make sure the results are interpreted correctly.
Response: We appreciate this important suggestion. We have addressed a possible selection bias of the control group in the Discussion section (Lines 262–263).
4. The finding of lower CD34+ cell yields in the salvage group is important. The manuscript should discuss whether DRD or lenalidomide specifically contributed to poor mobilization, and whether alternative mobilization strategies were considered.
Response: We appreciate this valuable suggestion. We have discussed the negative effects of lenalidomide and daratumumab on stem cell mobilization, the potential role of plerixafor, and a practical strategy for stem cell collection in the Discussion section (Lines 218–222).
5. The higher rate of hypogammaglobulinemia in the salvage group is notable, but the manuscript should provide more detail on infection monitoring, prophylaxis, and management, especially given the known infection risks with daratumumab.
Response: We appreciate this valuable suggestion. We have highlighted the importance of immunoglobulin replacement and monitoring serum IgG levels in the Discussion section (Lines 231–236).
Detailed Comments
Need more clarity on figures and tables:
Response: Thank you so much for your advice.
Figure 1:
Please label exclusion criteria and group assignments more clearly in the flow diagram, and indicate the number of patients at each step and in the final analysis.
Response: We have re-labeled and replaced the flow diagram to improve clarity.
Figure 2:
The legend should define all symbols and clarify abbreviations such as “EPd” and patient numbers.
Response: We have included comprehensive and self-explanatory legends, which indicate all symbols and abbreviations in the figure.
Figure 3 :
Axes and group labels need clearer definition. Specify in the legend which line represents each group, and, if possible, add the number at risk at each time point.
Response: We have included comprehensive and self-explanatory legends, which indicate axes and group labels in the figure. We also included the number at risk at the bottom of each time point.
Ensure all abbreviations (e.g., sCR, VGPR, MRD, TTNT, EPd) are defined at first use in both the main text and figure/table legends. For example, “EPd” is used in Figure 2 but not defined in the abbreviations list.
Response: We have checked that all abbreviations are now defined.
Consistently use standard abbreviations (e.g., sCR, VGPR, MRD) and define them at first use.
Response: We have checked that all abbreviations are now consistently used and defined at their first use.
Round 2
Reviewer 1 Report
The authors addressed all the comments
The authors addressed all the comments